# Structural Variations in Biobased Polyfurfuryl Alcohol Induced by Polymerization in Water

**DOI:** 10.3390/polym15071745

**Published:** 2023-03-31

**Authors:** Pierre Delliere, Antonio Pizzi, Nathanael Guigo

**Affiliations:** 1Institut de Chimie de Nice, Université Côte d’Azur, CNRS, UMR 7272, 06108 Nice, France; 2LERMAB-ENSTIB, University of Lorraine, 27 rue Philippe Seguin, 88000 Epinal, France

**Keywords:** Poly(furfuryl alcohol), furan resin, ring opening, NMR, carbonyls

## Abstract

Poly(furfuryl alcohol) is a thermostable biobased thermoset. The polymerization of furfuryl alcohol (FA) is sensitive to a number of side reactions, mainly the opening of the furan ring into carbonyl species. Such carbonyls can be used to introduce new properties into the PFA materials through derivatization. Hence, better understanding of the furan ring opening is required to develop new applications for PFA. This article studies the structural discrepancies between a PFA prepared in neat conditions versus a PFA prepared in aqueous conditions, i.e., with more carbonyls, through NMR and MALDI ToF. Overall, the PFA prepared in water exhibited a structure more heterogeneous than the PFA prepared in neat conditions. The presence of ketonic derivatives such as enols and ketals were highlighted in the case of the aqueous PFA. In this line, the addition of water at the beginning of the polymerization stimulated the production of aldehydes by a factor two. Finally, the PFA prepared in neat conditions showed terminal lactones instead of aldehydes.

## 1. Introduction

In recent decades, the need for high-performance, low-weight materials has increased. Good performance/density ratios can be achieved by using composites. Composite thermosetting materials have been highlighted as good candidates to meet the requirement of demanding applications needing high thermal properties and good mechanical properties as well chemical resistance [1].

Nowadays, the thermoset market is dominated by polyurethane, phenolic, amino, epoxy and polyester resins. The main applications of thermoset resins include paints, coatings, transportation, civil engineering, electronics and furniture [2]. Nonetheless, most of the thermoset resins commercially available are both toxic and fossil-based [3]. Phenol-formaldehyde resins, which are widely used in wood-adhesives and foundry binders, are a good example [4]. Concern over these compounds has led to research on potential replacements such as tannins [5,6] and lignins [7]. In this regard, the resins obtained from furfuryl alcohol (FA) are valuable alternatives. Indeed, FA can be combined with tannins [8,9], lignins [10,11], epoxidized linseed oil [12] or humins [13]. In addition, composites prepared from polyfurfuryl alcohol (PFA) resins and different fillers such as cork [14], plywood [15], kenaf fiber [16], cellulose [17], silica [18] and carbon nanoforms [19,20,21] were reported.

However, the thermoset resulting from the FA polymerization is rather brittle despite its good thermal properties [22]. Such brittleness arises from the dense structure of PFA. As a matter of fact, the chemical structure of PFA comprises a dense network of furfuryl units crosslinked together by Diels-Alder cycloadducts [23]. The polymerization behavior of FA is reported to depend on the initial conditions [24,25]. The polymerization is especially disturbed by the presence of protic and polar solvents [26,27]. Indeed, a PFA polymerized in aqueous solution has a lower *T*_g_ and crosslink density in comparison with a PFA prepared in neat conditions [27]. The disturbance induced by water is attributed to the furan ring-opening side reaction. Indeed, when placed in an aqueous medium, furfuryl alcohol tends to open into levulinic acid; thus, it was suggested that a similar reaction occurred in PFA [28,29]. 

The presence of carbonyl-based open structures within PFA has been reported numerous times in the past [30,31,32,33,34]. In recent research, the carbonyls resulting from the furan ring-opening were quantified [35], and the effects of the polymerization conditions on their formation were studied [36]. Subsequently, the carbonyls within PFA were exploited to tune the final properties of the material, i.e., reducing its brittleness. As a proof of concept, a flexible biobased amine (Priamine 1071) was able to react with the carbonyls of PFA resins to form Schiff bases. It is possible to change the macroscopic mechanical behavior of the PFA passing from a brittle material (without amine) to a progressively more ductile material when increasing the stoichiometric ratio of amine (vs. carbonyls) [37]. Yet, to fully develop the potential of PFAs’ carbonyls, a better understanding of their nature is required. Previous work focused on the carbonylated moieties found in a PFA synthetized in aqueous media. This study used 2D NMR to highlight the presence of various carbonyls, including aldehydes, in PFA [38]. 

The aim of the present article is to further apprehend the nature of the carbonyl groups in PFAs by comparing two types of PFA. A first one was prepared in neat conditions (PFA°), i.e., without additional water, and a second one was synthetized in 50% aqueous solution (PFA+). The PFAs’ structures were compared using 2D NMR techniques (HMBC and HSQC) and ^1^H NMR quantification. In addition, MALDI-ToF measurements were performed to further highlight the structural differences between the samples.

## 2. Materials and Methods

### 2.1. Materials

Dimethyl terephthalate (*qNMR*), furfuryl alcohol (96%w), maleic anhydride (99%w), hydroxylamine hydrochloride (99%w), triethanolamine (99%w), hydrochloric acid (37%w) and DMSO (98%) were purchased from Merck. VWR provided the ethanol (96%v). The d_6_-DMSO was purchased from Eurisotop. The water used in the synthesis and characterization was distilled using a SI Analytics distillation unit.

### 2.2. Synthesis of PFA Resins

Two types of PFA resins were prepared. The first one, called PFA°, was only composed of FA. The polymerization was initiated with 2%w of maleic anhydride and conducted in an oil bath. The temperature was kept at 85 °C for one hour, then 30 min at 100 °C and finally 110 °C until the rubbery state was reached. Over the course of the polymerization, a few samples were collected and characterized. This synthesis method was used to match the ones of previous work on PFA systems [26,27,35,36].

The second PFA, called PFA+, was synthesized by initiating a 50%w FA-water solution with 2%w of maleic anhydride, with respect to FA. Then, the same procedure as PFA° was applied.

### 2.3. Characterization of the PFA Resins

The conversion degree of the resins was determined using a DSC 1 from Mettler-Toledo. The conversion degree (α) was determined using the residual enthalpies as previously reported [24,25,26].

The total carbonyl content was determined using the oximation method based on the methods developed in the literature [35,39,40]. Briefly, the PFA resins were exposed to hydroxylamine hydrochloride (0.44 M) and triethanolamine (0.52 M) in ethanol/DMSO (1:2 vol) at 80 °C for 24 h. As a consequence, the carbonyls are converted in oximes, and HCL is freed. The triethanolamine in the medium captures the HCl, then the left-over base is titrated with 0.06 M HCl. A blank without PFA was also performed. Both the blank and quantifications were triplicated. The carbonyl content was calculated with Equation (1). In Equation (1), $C_{HCl}$ stands for the concentration of HCl, $m_{PFA}$ the mass of PFA, $V_{0}$ the volume of HCl required to titrate the blank and V the volume of HCl required to titrate the PFA solution.
(1)$$CO\;content\;\left(mmol/g\right)=C_{HCl}*\frac{V_{0}-V}{m_{PFA}}$$

To determine the conversion degree—α, 5–10 mg PFA resins were placed in high-pressure steel crucibles (30 µL). The samples underwent a temperature ramp from 0 to 300 °C at 2 °C/min. The resulting exothermic was then integrated and the conversion degree calculated using Equation (2). In Equation (2), ${\Delta H}_{residual}$ refers to the enthalpy of the exothermic peaks of pre-polymerized PFAs, while ${\Delta H}_{max}$ stands for the enthalpy of the unpolymerized solution.
(2)$$\alpha =\frac{{\Delta H}_{max}-{\Delta H}_{residual}}{{\Delta H}_{max}}$$

The Appendix A presents the data resulting from the C=O quantification and the conversion degrees determination.

### 2.4. Nuclear Magnetic Resonance (NMR)

All the NMR experiments were performed at 25 °C on a Bruker Advance DRX 500 spectrometer. The apparatus was equipped with a 5 mm PA DUL 500S2 C-H-D-05 Z probe. The operating frequencies were of 125.77 MHz and 500.13 MHz for ^13^C and ^1^H, respectively. The manufacturer supplied all the pulse sequences. To reference the spectra, the peak of DMSO was used at 39.52 ppm for the ^13^C and 2.50 ppm for the ^1^H. Masses of about 100 mg of PFA and 700 µL of d_6_-DMSO were used.

The ^13^C NMR experiments were performed using the “zgpg30” pulse sequence, a total of 20,000 scans halted by a relaxation delay of 2 s were used. The Heteronuclear Single Quantum Coherence (HSQC) spectra were acquired with the pulse sequence “hsqcedetgpsisp2.3”. For the HSQC, 54 scans were recorded, and a relaxation delay of 1.5 s was used. The Heteronuclear Multiple Bond Correlation (HMBC) experiments used the “hmbcetgpl3nd” pulse sequence with a relaxation delay of 2 s and the acquisition of 100 scans. Finally, the ^1^H NMR experiments were recorded with the “zg” pulse sequence with a relaxation delay of 30 s and an accumulation of 100 scans. The calculations used for the quantifications are detailed in previous work [38].

### 2.5. MALDI ToF Mass Spectrometry

The samples were treated with a NaCl solution (1.5 μL of a 0.1 M) in a methanol/water mixture (1:1) to increase ion formation, and a drop placed on the MALDI target (3 mm diameter) steel plate and dried. The samples and the matrix were then mixed in equal amounts, and 1.5 μL of the resulting slurry was placed on the MALDI target and dried at 40 °C for 2 h before being analyzed. A matrix of 2,5-dihydroxy benzoic acid was used. Red phosphorous (500–3000 Da) was used as reference for spectrum calibration. Finally, after evaporation of the solvent, the MALDI target was introduced into the spectrometer. The spectra were recorded on a KRATOS AXIMA Performance mass spectrometer from Shimadzu Biotech (Kratos Analytical Shimadzu Europe, Ltd., Manchester, UK). The irradiation source was a pulsed nitrogen laser with a wavelength of 337 nm. The length of one laser pulse was 3 ns. Measurements were carried out using the following conditions: polarity-positive, flight path-linear, 20 kV acceleration voltages, 100–150 pulses per spectrum. The delayed extraction technique was used applying delay times of 200–800 ns. The software MALDI-MS was used for the data treatment. The oligomers can appear in the spectra either corresponding to their molecular weight or to their molecular weight +23 Da of the Na^+^ ion derived from the NaCl used as enhancer. The spectra precision is of ±1 Da.

## 3. Results

This study aims at better understanding the effect of water on the polymerization of FA towards the structure of the resulting polymers. The focus was put on the side reactions occurring during aqueous and neat polymerization of FA.

First, HSQC experiments of PFA° and PFA+ resins were performed to highlight their main structural differences. For an easier comparison, resins with a conversion degree of about 0.80 were selected.

Figure 1 displays a magnification of the HSQC spectra of PFA+ and PFA° located on the methyl area. In Figure 1, PFA°(red) is stacked above PFA+ (blue). The opposite viewpoint in which the PFA+ HSQC spectrum is shown in Appendix A. This magnification highlights the main types of methyl found in PFA. Three groups with significatively different chemical environments can be identified in soluble PFA resins. 

The first group centered around 2.2/14 ppm can be attributed to the methylated furans as depicted in Figure 1. The methylated furans are formed during the conjugations of furanic chains through the loss of a hydride ion [23]. Such methyls are observed in both PFA° and PFA+. Nonetheless, a larger dispersion of chemical shift occurs in the case of PFA+, thus suggesting a higher heterogeneity of the furanic structures in PFA+ resins. Another distinct group of methyls at 1.95/15 ppm is observed only in the case of PFA+. This cluster does not spread as much as the main one. Thus, it might belong to a free molecule (e.g., dimers, trimers) rather than to a group linked to the main macromolecular chains. 

A second group around 2.0/30 ppm is observed. Previously, it had been attributed to the levulinic-like structures in Figure 1 [38]. This structure is believed to emerge from the end-chain hydrolytic ring-opening of furan units. Two other clusters of free molecules can be observed as well at 2.4/29 ppm and 2.25/28 ppm. The 2.4/29 ppm clusters are due to free levulinic acid [38]. The nature of the 2.25/28 ppm cluster remains unclear, however.

Finally, on the right-hand side of Figure 1, five discrete clusters of methyls can be identified for the PFA° resins, while three, much wider, clusters are observed in the case of the PFA+ resin. The ^1^H NMR spectra in Figure 2 can help to better understand their nature with a focus on the 1.1 to 2.4 ppm. The PFA+ spectrum exhibits three broad methyl signals. The one at 2.2 ppm been assigned to the methylated furans and is found as well in the PFA° spectrum. An even broader signal is observed from 1.90 to 2.15 ppm. It was assigned to the end-chain levulinic units. While the two PFAs are at similar conversion degrees, the 1.90–2.15 ppm signal is much broader in the case of PFA+ compared to PFA°. This is consistent with the higher C=O content in PFA+. 

Moreover, the PFA+ spectrum displays a low-intensity broad signal from 1.2 to 1.5 ppm. Such signal is not observed in the PFA° spectrum. Thus, it is obviously emerging from a water-induced side-reaction. The HMBC spectra in Appendix A presents the methyl correlations observable for both PFA° and PFA+. The broad methyl signal from 1.2 to 1.5 ppm of PFA° correlates mainly with a peak around 106 ppm and to a lesser extent with peaks around 32, 43 and 156 ppm. Due to the overlapping with furan and CH_2_ peaks, the nature of these methyls remains unclear.

Instead of the broad 1.2 to 1.5 ppm signal, the spectrum of PFA° rather displays four well-defined peaks at 1.24, 1.43, 1.63 and 1.75 ppm. The chemical environment behind these peaks will be discussed later in this article. Finally, both PFA° and PFA+ exhibit a methyl signal at 0.85 ppm, most likely another side-reaction product. The 0.85 ppm methyl peak was also identified in a previous study [41].

In the following section, the discrepancies between the CH_2_ groups of PFA° and PFA+ will be discussed.

Figure 3 displays a magnification of the HSQC spectra of PFA+ and PFA° centered in the CH_2_ region. In Figure 3, the spectrum of PFA° is stacked above the one of PFA+, while the inverse situation (i.e., PFA+ above PFA°) ° is shown in Appendix A.

In Figure 3, four main types of CH_2_ can be identified for both PFA+ and PFA°. The bottom-left one corresponds to furfuryl alcohol and its oligomers. The top-left group corresponds to the furfuryl units as shown in Figure 2 [42,43]. For these two clusters, the spectra of PFA+ and PFA° superimpose nicely, thus showing only a few structural divergences regarding these moieties. Yet, small clusters around 3.7/28 ppm are observed on the PFA+ spectrum.

The bottom part in Figure 3 encompasses clusters that would match with ether functions [44]. The clusters around 3.8/75 ppm are common to both PFA+ and PFA° and are most likely the result of head-to-head condensation of furfuryl alcohol molecules. As depicted in Figure 3, such condensation is susceptible to occur. Yet, it ultimately leads to furfuryl units by freeing formaldehyde [42]. A few other clusters are also visible on the PFA+ spectrum around 4.2/72 ppm, as well as larger ones around 3.5/68 ppm and a small one at 3.9/62 ppm for PFA°. The exact nature of the molecules causing these correlations is, however, difficult to point out.

The top-right group in Figure 3 displays CH_2_ clusters that have been linked to carbonyl-based open structures [38]. A very wide cluster 2.3–3.3/35–45 ppm is observed in the PFA+ spectrum. In previous work, this cluster was assigned to the open structure of Figure 4 and/or to the end-chain levulinic units of Figure 1. In the case of PFA°, a similar, yet much narrower, cluster is observed.

The carbonyl content of the studied PFA° is about 2.0 mmol/g, while PFA+ contains about 3.0 mmol/g of carbonyls. As a consequence, the intensity of the CH_2_ neighboring the carbonyls in PFA+ should be higher than in PFA°. In addition, a wider distribution of chemical shifts is expected due to the chemical environment of CH_2_ in PFA+. The intensity and area of the ^1^H signals from the CH_2_ cannot be compared accurately due to a lack of proper baseline. Nonetheless, the HSQC spectra in Figure 3 displays a much higher heterogeneity of the CH_2_ comprised between 2.2 and 3.3 ppm for PFA+, in comparison with PFA°. On top of that, the PFA+ spectrum displays clusters, mainly around 3.8/41 ppm, 3.3/40 ppm, 2,8/42 ppm and 3.2/43 ppm, which are not found in PFA°. Thus, introducing water in the initial reaction medium induces ring-opening reactions which do not occur in a neat system. Finally, the PFA° spectrum in Figure 3 displays CH clusters around 3.2/47, 3.4/50 ppm not visible on the PFA+ spectrum. Hence, the side reactions occurring in a neat polymerization are of another nature than the ones occurring in aqueous systems. These sets of side reactions might explain the influence of the reaction media on the polymerization of PFA [24,25,26].

In the following section, the structural differences in the C=C and C-O areas will now be discussed.

Figure 4 depicts partial HSQC spectra of PFA° and PFA+ with PFA° on top of PFA+. The opposite situation with PFA+ on top of PFA° is available in Appendix A. In Figure 4, three groups of clusters can be isolated. 

The top-left group corresponds to the C3 and C4 of furanic groups found in PFA, i.e., furfuryl units of Figure 2 and FA [42]. As previously observed in Figure 3, the spectra of PFA+ and PFA° superimpose nicely for the furanic moieties.

The bottom-left group encompasses clusters matching with alkene groups [44]. They have been attributed to the conjugated structure of Figure 5 and its Diels-Alder adduct. The mechanisms leading to the conjugated structure and the Diels-Alder crosslinking were investigated in the past [42,45]. In Figure 5, the HC=C clusters at 6.3/137 ppm, 6.0/133 ppm and 5.8/130 ppm are common to both PFA systems, thus suggesting an assignment to the structures in Figure 5 and to the left-over of maleic acid remaining in the system. However, the PFA+ spectrum exhibits numerous clusters not observed in the case of PFA°. They might come from enolized carbonyl functions [38].

Indeed, as depicted in the top-right corner of Figure 4, the abundance of clusters is characteristic of the PFA+ spectrum. Their chemical shifts are consistent with either enols or hydrated carbonyls [44]. This hypothesis is based on the fact that PFA° has very few clusters in this area, which is consistent with a lower carbonyl content. Moreover, recent research on the quantification of carbonyls within PFA systems showed that longer derivatization times are necessary to fully quantify the carbonyls [35] in PFA in comparison, for instance, with lignins [39] and bio-oils [40]. As a matter of fact, the reactivity of carbonyls in PFA resins can be used to tune the properties of the final materials [37]. Hence, the means of displacing the enol–carbonyl and/or ketal–carbonyl equilibria can be used to fully exploit the potential of the carbonyls within PFA systems.

In a recent study, the presence of aldehyde groups in PFA+ was highlighted, but no comparison was made with PFA° [38].

Figure 5 displays the HSQC spectra of PFA+ and PFA° focusing on the aldehyde region, with the spectrum of PFA° on top of the one of PFA+. The opposite situation is available in Appendix A.

In Figure 5, two populations of aldehydes can be identified in the PFA° system. A first one below 180 ppm can be identified as furfural and similar groups. Indeed, the chemical shift of the aldehyde group in furfural is around 178 ppm, while other aldehydes are usually above 185 ppm [44]. It indicates that both PFA° and PFA+ bear aldehyde groups similar to the aldehyde of furfural.

Around 9.5/194 ppm, an aldehyde cluster is observed in both spectra. A previous study identified this cluster, depicted in Figure 6, as the result of the oxidative ring-opening of terminal furan units. It is believed that this conjugated aldehyde reacts through a Diels-Alder reaction, with linear furans ultimately leading to a stiffening at the surface of the resin. 

Finally, a third set of aldehydes is observed around 200 pm for PFA+ only. It suggests that the presence of additional water induces the formation of aldehydes during FA’s polymerization. To further investigate the influence of water on the formation of aldehydes in PFA resins, they were quantified in PFA° resins employing the same methods ^1^H NMR as previously on PFA+ [38]. Two methods were employed. The first one used dimethylterephtalate as an external standard. The second method employed the left-over furans as reference. This last method approximates the macromolecular network as a repetition of furfuryl unit. However, it does not require the sample mass for the calculations [38]. The results are compiled in Table 1 and illustrated with the ^1^H quantitative NMR spectra in Figure 6.

It was found that the concentration of aldehydes in PFA+ increase with the conversion degree, as well as the aldehyde ratio—over all the carbonyls [44]. Table 1 reports a similar behavior for PFA°. With the method using an external standard, the aldehyde concentration increases very slightly over time with the course of polymerization. Nonetheless, the aldehyde ratio remains steady during the whole polymerization of FA in the neat system. A similar trend is observed with the left-over furan method. Overall, PFA° comprised up to 0.13 mmol/g of aldehydes, i.e., about 7% of its carbonyls. In comparison, the PFA+ resins comprise about 0.50 mmol/g of aldehydes in resin with a conversion degree of 0.83 and about 0.20 mmol/g of aldehydes for a resin with a conversion degree of 0.47. Consequently, polymerizing FA in a 50% aqueous solution stimulates the production of aldehydes either from a water-based or an oxidative pathway.

To further emphasize the structural discrepancies between PFA° and PFA+, the ester functions will now be discussed.

In the past, the existence of ester functions in a PFA network was mentioned numerous times [22,27,33,34,35,36,46]. The lactone structure in Figure 7 was proposed as the source of these esters. In the following, the HMBC method is used to investigate their nature.

Figure 7 displays the magnified HMBC spectrum of PFA°. As the concentration of esters is low, and due to an overlapping with other signals, only a few insights can be given. Figure 7 shows ^2^J and ^3^J correlations which match well with the predicted chemical shifts of the lactone depicted in Figure 7. The predicted chemical shifts were obtained from the NMR predict software from University of Lausanne “NMR DB” and are available in Appendix A [47].

The nature of the other methyl peaks in PFA° cannot be elucidated as easily. However, the peaks at 1.91, 1,75, 1,48 and 1,43 ppm all correlate with carbon peaks in the ester region, i.e., from 150 to 185 ppm. The PFA+ HMBC spectrum in Appendix A displays the 1.48/156 ppm and 1.91/172 ppm clusters as well. Nonetheless, the lactone is not observed in the PFA+ spectrum. This is consistent with previous work, including with the FTIR spectrum of PFA+ resins that exhibit the typical ester bands [27,35,36].

Therefore, some of the esters in PFA° are terminal lactones. The end-chain location of the lactones might explain why fewer aldehydes are present in the PFA° compared to PFA+. The presence of furfuryl levulinates is also possible, although their signals might be overlapping with the ones of levulinic acid.

From the comparison of the MALDI ToF spectra in Figure 8A–D and Figure 9A–D and Appendix A, there are more open structures and ketone structures in PFA+ than in PFA°. This can be seen from the greatest abundance of the structures at 326–327 Da, 334 Da and 574–576 Da (Figure 8).

The 326–327 Da is in line with the open structure already found in the NMR section, and shown in the simpler structure corresponding to the MALDI peak at 198 Da. As shown below, the 334 Da and 574–576 Da ones are also examples of oligomers which includes some open ketone structure. 

The two structures at 334 Da and 574–576 Da are also examples of how the Diels–Alder rearrangement of furanic chains participates in higher molecular weight oligomers.

For the 325 Da, the peak is much more intense in relation to the rest of the spectrum, and for the 334 Da, while the size is similar for the two PFA resins. PFA+ presents two peaks of equal height that indicate that the abundance of the structure is higher. The ketone-carrying structure at 574–576 Da also has a much more pronounced peak for PFA+, and all its multiples are far more intense than in PFA°. The ketone structures at 360 Da and at 440 Da also present peaks clearly much more intense in PFA+. The structure at 361 Da (including the 23 Da for Na^+^) is also one example of the structure determined in the NMR section (Figure 7).

There are other indications of this. The peak at 152 Da in the PFA+ is smaller than the 154 Da one in PFA°, while in the latter the closer non-ketone form is instead more evident at 157 Da.

There are several structures in which a charged atom is still present. One example is the peak at 161 Da that could be interpreted as any one of the two structures of Figure 9.

However, both might be effectively present, as well as being fabricated in the MALDI ToF experience itself. Yet, the presence of several other similarly charged structures at 176 Da, 240–241 Da, 481–483 Da and 496 Da appear to indicate that these structures are not just a fabrication of the MALDI spectrometer (Figure 10). 

The peak at 176 Da, which is much stronger in PFA°, can be ascribed to two different structures contributing to the same peak (Figure 11). The much smaller residual 176 Da in PFA+ might definitely be due to the open ketone structure.

With the second one having been identified in the NMR investigation. Related to these forms are the peaks at 190 Da and 194 Da, assigned to the structures in Figure 12.

At 190–191 Da, with the second one is far more likely than the first one, and the form at 194 Da (Figure 13). 

Also interesting is the behavior of the 241–242 peak. This can be attributed to two structures in Figure 14, both being likely.

Among the structures observed is also one structure as identified in the NMR analysis in which the terminal unit of an furanic oligomer is a conjugated aldehyde resulting from the end-chain oxidative ring-opening of furans in PFA resins (Figure 15). This is observed at 404 Da –no Na^+^ and 428 Da (with Na^+^, presenting the structure in Figure 15)

Finally, the structure formed by Diels-Alder adducts are well present in the PFA resins examined, with the peaks at 310 Da, 318 Da, 334 Da (with Na^+^) again showing more open forms in PFA+, 456 Da (no Na^+^) and 476–478 Da (with Na^+^), 470 Da (no Na^+^) and 493 Da (with Na^+^), 481–483 Da and 496 Da, and 574–575 Da (Appendix A) such as the species at 456 Da and 470 Da, as follows in Figure 16.

## 4. Conclusions

In this article, the chemical structures of two types of PFA resins were investigated. The first PFA was prepared in neat conditions (PFA°), while the second one was prepared in a 50%w aqueous solution (PFA+). To investigate the structures, both NMR and MALDI ToF were employed. Overall, the study highlights a higher heterogeneity in PFA+ as exemplified by broader HSQC clusters. Such heterogeneities were attributed to water-induced side reactions. They involve the formation of ketonic and aldehyde species. Moreover, it was shown that the resulting ketones are sensitive to enolization and hydration. On the other hand, PFA° displays fewer ketone derivatives. In addition, the presence of aldehyde groups are about two times more important in PFA+ compared to PFA°, thus indicating that polymerization in aqueous conditions favors their formation. Furthermore, the presence of lactone end-groups in PFA° was confirmed, and it somehow explains the lower concentration of aldehydes for PFA°. Finally, the results from the MALDI ToF investigations corroborated the higher abundance of open structures in PFA+ systems and thus helped to propose some potential structures.

## Data Availability

Publicly available datasets were analyzed in this study. These data can be found here: https://hal.science/ (accessed on 30 March 2023).

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
