# Peer review of "Structural Variations in Biobased Polyfurfuryl Alcohol Induced by Polymerization in Water"

_polymers, 2023, doi:10.3390/polym15071745_

Round 1

Reviewer 1 Report

This submission details the effect of water addition on furfuryl alcohol polymerization and cross-linking reaction. This is relevant in the context of formulating bio-based/lignin-derived products for targeted applications. The molecular structure of such crosslinked resins would control their properties and this study thus elucidates two ways of making such resins and how these different synthesis routes lead to diversity in product structures. Using NMR and MALDI-based routes, the authors propose several structures in their cross-linked products. 

I found the manuscript clearly written and well-structured. I feel that certain points regarding experimental design can be made clearer.

Here are some suggestions/questions for the authors:

1. Can the authors comment on the immediate applications of their findings? Does the presence of more open structures/aldehydes in the water-derived structures lead to an observed increase in mechanical properties at room temp (through H bonding interactions)? I would recommend some sort of property characterization of the two class of thermosets to make a stronger case for caring about this properties. If this work has already been done before, can the authors please highlight this reference. 

2. Line 65 - techniques has been mispelled

3. Line 86 - synthesized

4. Line 97 - as carbonyl group determination is one of the core objectives of the study, please expand more on how the oximation was done and what the results of the oximation are. Have the oximation and the DSC results been reported in the ESI? please direct the reader to these results and summarize key findings in the main text discussion.

5. Was the initiator removed? or does it get incorporated into the resin? Can the initiator show up in the subsequent NMR investigations. 

6. As a note on readability, one figure with two possible reaction mechanisms/structures to differentiate the two PFA0 and PFA+ can be included. I leave this up to the authors' decision.

7. Schemes 12-15 could be more descriptive with details on which structure corresponds to which product. i.e. left is PFA0 and right is PFA+, etc.

8. Line 485 - the should be omitted or replaced with thus

Author Response

Dear reviewer,

We would like to thank you for your time and the meaningful comments that you provided. Your constructive comments will strongly help to improve the quality of the manuscript. Please find below our point by point answers to your queries:

1/Indeed, this issue was already adressed in a previous paper and we have added the reference to this work.

2/ Thanks - this has been changed

3/ Thanks - The word has been written correctly

4/ Indeed, this is an important point. We have added the details on the oximation method in the materials and methods section (reagent for oximation, detailed protocol and equation) as well as the detailed protocol for the determination of the conversion degree by DSC. The modifications will appear mainly between line 102 and 124. Table S1 was added in ESI to present the main results from the oximation and conversion degree.

5/ Yes the initiator, maleic anhydride - rather in the form of maleic acid - remains trapped into the material during polymerization so that it could contribute to some NMR signals. We have added this in the comment of Figure 5 (line 309).

6/ This indeed would have been interesting. Based on this remark, we have tried to make such scheme but, in the end, too much features should appear in this scheme so that it became too complex and confusing. We have decided to keep the initial version where only short schemes are highlighted. 

7/Thanks - indeed most of them are connected to both PFA° and PFA+ but in some cases (like scheme 11 and 12) some of them are more specific.

8/ Thanks. Indeed we have changed the with thus!

Reviewer 2 Report

Authors developed PFA resins through different methods: in neat and aqueous conditions. The molecular structure of the PFAs was elucidated, the main differences were discussed. The manuscript is well written and the results were well discussed. Even the limitations of the analysis were considered in the discussion.

The manuscript can be accepted after minor revisions/suggestions:

1. DSC and  oximation method (FTIR???) are described in "Materials and Methods", but the results are not presented/discussed. Authors should (at least)  add these results to supplementary materials;

2. Authors should elucidate (Introduction), how carbonyl groups in PFA can tune the properties of final thermoset material, based on reference 37.

Author Response

Dear reviewer,

We would like to thank you for your time and the meaningful comments that you provided. Your constructive comments will strongly help to improve the quality of the manuscript. Please find below our point by point answers to your queries:

1/ Thanks - based on your remark we have added in the materials and methods section the detailed protocol for oximation and the determination of the conversion degree based on DSC experiments. Results were added in Table S1 in ESI. The changes are located between line 102 and 124.

2/ Thanks - we have elaborated a bit more on the interest of tuning PFA properties through the formation of Schiff bases with the carbonyls contained in PFA by utilizing a flexible amine (Priamine 1071). You can see the changes in the introduction between line 58 and 63.